# Advances in the Structure of GGGGCC Repeat RNA Sequence and Its Interaction with Small Molecules and Protein Partners

**DOI:** 10.3390/molecules28155801

**Published:** 2023-08-01

**Authors:** Xiaole Liu, Xinyue Zhao, Jinhan He, Sishi Wang, Xinfei Shen, Qingfeng Liu, Shenlin Wang

**Affiliations:** 1State Key Laboratory of Bioreactor Engineering, East China University of Science and Technology, Shanghai 200237, China; y85210065@mail.ecust.edu.cn (X.L.); 20000624@mail.ecust.edu.cn (X.Z.); y85220142@mail.ecust.edu.cn (J.H.); 20000612@mail.ecust.edu.cn (S.W.); 20000603@mail.ecust.edu.cn (X.S.); 20000608@mail.ecust.edu.cn (Q.L.); 2Beijing NMR Center, Peking University, Beijing 100087, China

**Keywords:** amyotrophic lateral sclerosis, frontotemporal dementia, *C9orf72*, GGGGCC, G4

## Abstract

The aberrant expansion of GGGGCC hexanucleotide repeats within the first intron of the *C9orf72* gene represent the predominant genetic etiology underlying amyotrophic lateral sclerosis (ALS) and frontal temporal dementia (FTD). The transcribed r(GGGGCC)_n_ RNA repeats form RNA foci, which recruit RNA binding proteins and impede their normal cellular functions, ultimately resulting in fatal neurodegenerative disorders. Furthermore, the non-canonical translation of the r(GGGGCC)_n_ sequence can generate dipeptide repeats, which have been postulated as pathological causes. Comprehensive structural analyses of r(GGGGCC)_n_ have unveiled its polymorphic nature, exhibiting the propensity to adopt dimeric, hairpin, or G-quadruplex conformations, all of which possess the capacity to interact with RNA binding proteins. Small molecules capable of binding to r(GGGGCC)_n_ have been discovered and proposed as potential lead compounds for the treatment of ALS and FTD. Some of these molecules function in preventing RNA–protein interactions or impeding the phase transition of r(GGGGCC)_n_. In this review, we present a comprehensive summary of the recent advancements in the structural characterization of r(GGGGCC)_n_, its propensity to form RNA foci, and its interactions with small molecules and proteins. Specifically, we emphasize the structural diversity of r(GGGGCC)_n_ and its influence on partner binding. Given the crucial role of r(GGGGCC)_n_ in the pathogenesis of ALS and FTD, the primary objective of this review is to facilitate the development of therapeutic interventions targeting r(GGGGCC)_n_ RNA.

## 1. Introduction

Amyotrophic lateral sclerosis (ALS) and frontal temporal dementia (FTD) are two neurodegenerative disorders characterized by progressive degeneration and dysfunction of neuronal architecture [1,2,3,4,5]. Both diseases have a fatality rate typically occurring within three to five years after the onset of symptoms [6,7]. ALS, affecting approximately two individuals per 100,000, is characterized by the degeneration of motor neurons, leading to muscle weakness and atrophy [8,9]. FTD, the second most prevalent form of dementia in individuals under the age of 65, is typically characterized by atrophy of the frontal and/or temporal lobes, manifesting as heterogeneous symptoms encompassing behavioral changes (behavioral variant FTD, bvFTD), language impairment (primary progressive aphasia, PPA), or deterioration in motor skills [10]. Despite considerable efforts, the development of efficacious therapeutic strategies for the treatment of ALS and FTD remains a challenge [11].

The etiologies of ALS and FTD are various. Sporadic ALS (sALS) accounts for 90% of the ALS patients. The remaining 10% of ALS patients are familial ALS (fALS) caused by mutations. The fALS can be caused by the dysfunction of mutated proteins, such as SOD1 mutations, FUS/TLS, and TDP-43 provoked by *TARDBP* mutations [12,13], which lead to neurotoxicity. The aberrant elongation of the hexanucleotide repeat GGGGCC in the non-coding region of *C9orf72* has also been demonstrated to be causally associated with a ALS and FTD [8,14,15,16,17,18,19,20,21,22]. Aberrant expansion of the GGGGCC repeats is observed in 8% of sALS patients, as well as in more than 40% of fALS cases [14]. Individuals affected by ALS exhibit an average repeat count ranging from 700 to 1600, whereas healthy individuals possess fewer than 25 repeats [23,24,25,26]. As for FTD, approximately one-third of FTDs are familial, with autosomal dominant mutations in three genes accounting for the majority of inheritance, including progranulin (GRN), *C9orf72*, and microtubule- associated protein tau (MAPT) [27]. The co-occurrence of these two disorders among families provides support for their genetic linkage [28].

Moreover, aberrant expansion of short nucleotide repeats has been observed in many neurodegenerative diseases. CTG triplet amplification in 3′-UTR may occur in dystrophia myotonica protein kinase (DMPK) gene, and alternative splicing of junctophilin (JPH) gene exon 2a and ataxin8 (ATXN8) gene, which can, respectively, result in Muscular dystrophy type 1 (DM1), Huntington disease-like 2 (HDL2), and Spinocerebellar Ataxia 8 (SCA8). The amplification of CGG triplet in 5′-UTR of the fragile X mental retardation 1 (FMR1) gene may lead to Fragile X disorders (FXTAS), while (CAG)n in the exon of ataxin3 (ATXN3) may cause Spinocerebellar Ataxia 3 (SCA3). In addition, (ATTCT)n within an intron of the ataxin 10 (ATXN10) gene and (CCTG)n in the first intron of the zinc finger protein 9 (ZNF9)gene may lead to Spinocerebellar Ataxia 10 (SAC10) and Muscular dystrophy type 2 (DM2), respectively [29]. Characterization of pathological mechanisms by which these short nucleotide repeats cause fatal diseases has been the research focus aiming in finding the treatments.

Three mechanisms have been proposed to elucidate the pathological underpinnings of aberrant GGGGCC expansion (Figure 1) [30]. Firstly, these abnormal expansions can lead to a gain or loss of function in the associated gene [7,31]. Secondly, the transcribed r(GGGGCC)_n_ RNA forms RNA foci that recruit RNA binding proteins (RBPs), consequently impairing protein function and ultimately triggering intracellular cytotoxicity [7,32,33,34,35,36]. Lastly, non-ATG translation of the r(GGGGCC)_n_ sequence produces dipeptide repeats (DPRs) that exert neurotoxic effects within the central nervous system [7,37,38,39,40,41,42]. Among the three proposed mechanisms, the formation of RNA foci and recruitment of RBPs have garnered the most attention. This process involves the spontaneous liquid–liquid separation of r(GGGGCC)_n_, followed by a sol-gel phase transition by increased interactions [43]. The RNA foci can recruit various RBPs, including hnRNP H [44], Zfp106 [45], ADARB2 [46], Purα [47,48], and FUS [49,50], ultimately leading to disruptions in the intracellular environment [43,51]. The aberrant phase separation and spread of hnRNP H within r(GGGGCC)_n_ in ALS patients are key features in the pathogenesis of the disease [52].

Consequently, the investigation of the r(GGGGCC)_n_ RNA repeats structures and the interactions between r(GGGGCC)_n_ and small molecules is currently a highly prominent area of research. The primary objective is to identify lead compounds for the treatment of FTD and ALS [53]. The r(GGGGCC)_n_ sequence exhibits characteristic structural polymeric and has the ability to adopt either a hairpin or G-quadruplex (G4) structure [54]. Several small molecules have been discovered to possess strong binding affinity for r(GGGGCC)_n_. A majority of these compounds incorporate polyaromatic ring conjugated systems that effectively stabilize the G4 structures of r(GGGGCC)_n_, thereby inhibiting phase separation, disrupting protein–RNA interactions [55] and/or preventing non-ATG translation of DPRs [53]. Additionally, two drugs, riluzole and edaravone, have received approval for the treatment of ALS [56]. While these drugs can delay disease progression, they do not specifically target the r(GGGGCC)_n_ RNA [56]. Moreover, recent research by Meijboom and colleagues used the adeno-associated virus vector system to deliver CRISPR/Cas9 gene editing system into neuron cells, and successfully removed the hexanuclear repeat expansion from the *C9orf72* gene in the mouse model (500–600 repeats), as well as the patient-derived Induced Pluripotent Stem Cell (iPSC) motor neuron and brain organoid (450 repeats). This led to a reduction in RNA foci, DPRs and haploinsufficiency, major hallmarks of C9-ALS/FTD, making this a promising therapeutic approach to ALS/FTD diseases [57].

This review provides a comprehensive overview of the recent progress made in understanding the structures of r(GGGGCC)_n_, and the interactions between r(GGGGCC)_n_ and small molecules and between r(GGGGCC)_n_ and protein partners. We focus on elucidating the structural diversity of r(GGGGCC)_n_ and its implications for partner binding. Given the crucial role of r(GGGGCC)_n_ in the pathogenesis of ALS and FTD, the primary objective of this review is to support the development of drugs targeting r(GGGGCC)_n_ RNA.

## 2. The Structure of r(GGGGCC)_n_ RNA Repeats and the RNA within the RNA Foci of r(GGGGCC)_n_

### 2.1. The Solution Structures of r(GGGGCC)_n_ RNA

The r(GGGGCC)_n_ RNA is a guanine-rich sequence, which promotes the formation of G4 structures. While the tertiary structure of r(GGGGCC)_n_ remains to be fully elucidated, the secondary structures have been extensively studied. Circular Dichroism (CD) spectra are commonly used to demonstrate the G4 structures and the topology of G4, which for instance, the spectral patterns would provide the evidence of parallel, antiparallel, or other types of topologies (Figure 2). Nuclear Magnetic Resonance (NMR) spectra are able to provide more structural details, some of which yield full structural determination or provide evidence of co-existence of different conformations (Figure 2a). Depending on the sequence length and solution conditions, r(GGGGCC)_n_ can adopt different secondary structures, including G4 [58,59] and hairpin conformations [60]. In 2012, Adrian M. Isaacs and colleagues demonstrated that r(GGGGCC)_3_GGGGC can fold into G4 or double-stranded structures, with the topology being influenced by the presence of cation ions in the solution [61]. In a K^+^ buffer, it forms a stable parallel intramolecular G4 structure, while it becomes less stable in Na^+^ and Li^+^ solutions.

Further investigations by Pearson and colleagues employed circular dichroism (CD) spectroscopy (Figure 2a) and gel-shift assays, revealing that r(GGGGCC)_n_ (n = 2, −5, −6, and −8) predominantly adopt highly stable uni- and multi-molecular parallel G4 structures [62]. The abundance of G4 structures is influenced by the repeat number and RNA concentrations, with the proportion of multi-molecular G4 structures increasing as the number of repetitions rises.

The equilibrium between G4 and hairpin structures has also been observed in r(GGGGCC)_n_. In the absence of K^+^ ions, r(GGGGCC)_4_ RNA forms a hairpin conformation [62], featuring single-stranded bulges within the RNA chain. However, in a K^+^ buffer (Figure 2c), it adopts a parallel G4 structures (Figure 2g) [59]. This equilibrium between hairpin and G4 structures is suggested to be linked to the presence of an abortive transcript containing hexanucleotide repeats [55]. The G4 structure may hinder the transcription of full-length RNA and recruit RBPs in cells, contributing to disease pathogenesis. The equilibrium is biased towards the hairpin conformation with a higher repeat number of r(GGGGCC)_n_. Specifically, r(GGGGCC)_4_ predominantly adopts a G4 topology, while r(GGGGCC)_8_ RNA exhibits both G4 and hairpin structures, even in a K^+^ buffer, as confirmed by various biophysical methods. However, in a Na^+^ buffer, r(GGGGCC)_8_ RNA solely adopts a hairpin structure [55]. Furthermore, r(GGGGCC)_4_ undergoes a monomer-dimer equilibrium in a pH-dependent manner. At pH 6.0 and 25 °C, it exists as both a homodimer and a hairpin structure. Decreasing the temperature increases the population of dimeric RNA, which exhibits distinct structural differences compared to G4 structures in the presence of K^+^ [63]. Conversely, at neutral pH, r(GGGGCC)_4_ primarily adopts a hairpin conformation.

### 2.2. Structure of d(GGGGCC)_n_ DNA

High-resolution structures of d(GGGGCC)_n_ have been successfully determined [64,65]. Janez Plavec and colleagues utilized NMR spectroscopy to elucidate the structure of d[(GGGGCC)_3_GG^Br^GG] (represented by PDB codes 2N2D) [66]. The incorporation of a bromine-substituted guanine residue (G^Br^) contributed to the stabilization of the conformation, leading to a more rigid structure amenable to structural analysis. The d[(GGGCCC)_3_GG^Br^GG] sequence adopted an antiparallel G4 topology (Figure 2b) [67].

In 2015, Guang Zhu and colleagues employed CD, NMR, and native polyacrylamide gel electrophoresis (PAGE) to investigate the structures of d(GGGGCC)_n_ repeats. Their studies revealed distinct G4 folding patterns in the presence of K^+^ ions. Notably, d(GGGGCC)GGGG, d(GGGGCC)_2_, and d(GGGGCC)_3_ did not exhibit stable G4 structures. Instead, d(GGGGCC)_2_ and d(GGGGCC)_3_ displayed mixed forms of parallel and antiparallel G4 folding. On the other hand, d(GGGGCC)_4_ and d(GGGGCC)_5_ formed stable G4 structures. Specifically, d(GGGGCC)_5_ exhibited a combination of parallel and antiparallel G4 folds, while d(GGGGCC)_4_ adopted a homogeneous monomeric form characterized by a chair-type G4 structure [10]. In 2021, this group determined the crystal structure of d(GGGGCC)_2_ in both Ba^2+^ and K^+^ solutions, revealing an eight-layer parallel G4 structure for d(GGGGCC)_2_ (represented by PDB codes 7ECF and 7ECG) (Figure 2e) [68]. Jiou Wang and colleagues (Figure 2d) also confirmed that d(GGGGCC)_4_ adopts an antiparallel G4 (Figure 2f) [59].

**Figure 2 molecules-28-05801-f002:**
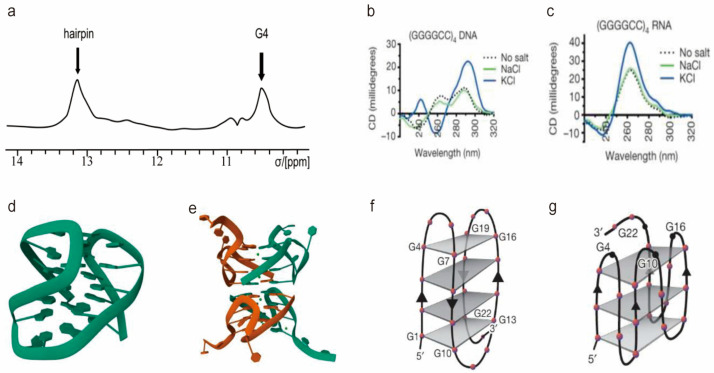
Biophysical methods in studying the r(GGGGCC)_n_ and the d(GGGGCC)_n_ structures. (**a**) ^1^H NMR spectra of r(GGGGCC), with highlighting the cross-peaks of the hairpin and G4 structures. (**b**,**c**) The CD spectra with the characteristic antiparallel G4 topology of d(GGGGCC)_4_ (**b**) and the parallel G4 for the r(GGGGCC)_4_ in the presence of 100 mM KCl. The ions dependent of G4 structures were shown. (**d**,**e**) The high-resolution crystal structures of d[(GGGGCC)_3_GG^Br^GG] (**d**) and d(GGGGCC)_2_ (**e**). (**f**,**g**) The proposed topology for the antiparallel DNA G4 formed by (GGGGCC)_4_ (**f**) and the parallel G4 topology formed by the r(GGGGCC)_4_ RNA (**g**). The (**b**,**c**,**f**,**g**) were reprinted from the reference [59]. (**d**,**e**) were reprinted from the reference [67,68], respectively.

### 2.3. Biological Phase Separation and Transition of r(GGGGCC)_n_

Biological liquid–liquid phase separation is a widely observed phenomenon in cells and plays a critical role in the formation of membraneless organelles, signal transduction, and DNA packaging [69,70,71,72]. As the strength of interactions in phase separation systems increases, a transition from a liquid to a solid state often occurs, resulting in the formation of insoluble gel-like states, many of which are associated with diseases [73]. Jain and colleagues demonstrated that r(GGGGCC)_n_ can undergo phase separation both in vivo and in vitro [43]. They found that phase separation of r(GGGGCC)_n_ occurs once a specific threshold of repeat value is reached, leading to a solution-gel phase transition as the strength of multi-base interactions increases (Figure 3). The formation of RNA foci is dependent on solution conditions and is reinforced by Mg^2+^ but impaired by monovalent cations such as K^+^ or Na^+^. The authors proposed that inter-chain hydrogen bonds stabilize intermolecular G4s, which serve as the building blocks of RNA foci. However, direct evidence of the secondary structure of r(GGGGCC)_n_ within RNA foci is still lacking.

Christopher E. Shaw and colleagues discovered that r(GGGGCC)_n_ RNA foci were detected in neuronal cell lines and zebrafish embryos expressing 38 or 72 repeats but not in those expressing 8 repeats [6]. This finding indicates that longer r(GGGGCC)_n_ sequences lead to nuclear retention of transcripts and the formation of RNA foci, which are resistant to the enzyme ribonuclease (RNase) [6,52]. Extended r(GGGGCC)_n_ sequences exhibit significant neurotoxicity and bind to hnRNP H and other RBPs. RNA toxicity and sequestration of RBPs may impair RNA processing and contribute to neurodegenerative diseases. In a study conducted by Simon Alberti and colleagues, it was demonstrated that RNA plays a crucial role in regulating the phase behavior of prion-like RBPs [74]. Lower RNA to protein ratios promote the separation of RBPs into liquid droplets, whereas higher ratios prevent droplet formation in vitro. When nuclear RNA levels are reduced or RNA binding is genetically ablated, excessive phase separation occurs, leading to the formation of cytotoxic solid-like assemblies in cells. The researchers proposed that the nucleus functions as a buffered system, with high RNA concentrations maintaining RBPs in a soluble state. Disruptions in RNA levels or the RNA binding abilities of RBPs result in abnormal phase transitions [75].

## 3. Disease Related RBPs That Bind to r(GGGGCC)_n_

### 3.1. hnRNP H and TDP-43

Heterogeneous nuclear ribonucleoprotein H (hnRNP H) is a member of the hnRNP family and functions as a multifunctional RBP involved in mRNA maturation at various stages [76]. It contains a modular domain consisting of tandem quasi-RNA recognition motifs (HqRRM1,2) at the N-terminus and a third qRRM3 at the C-terminus, situated between two glycine-rich segments [44,77,78]. The hnRNP H has the ability to bind G-rich RNA sequences containing at least three consecutive guanines [44]. In the brain cells of ALS patients, hnRNP H has been found associated with insoluble aggregation of r(GGGGCC)_n_, leading to aberrant alternative splicing [52]. This phenotype has been utilized as a biomarker for disease diagnosis. Furthermore, ALS/FTD patients exhibit splicing alterations in several key targets and insoluble hnRNP H, indicating that modifications along this axis are critical aspects of disease etiology [52].

James L. Manley and colleagues demonstrated that hnRNP H binds to r(GGGGCC)_n_ in vitro, and this interaction is dependent on the formation of G4s. The hnRNP H colocalizes with G4 aggregates in C9 patient-derived fibroblasts and astrocytes, but not in control cells, as proven by imaging on BG4, a G4 structure-specific antibody (Figure 4) [79]. Another study by Donald C. Rio and colleagues revealed that in sporadic ALS/FTD patients, insolubility of hnRNP H was associated with altered splicing of a wide range of targets [52]. Numerous ALS/FTD brains show high levels of insoluble hnRNP H sequestered in r(GGGGCC)_4_ RNA foci, resulting from RNA splicing defects involving intron retention [52]. These findings highlight previously unreported splicing abnormalities in extremely insoluble hnRNP H-related ALS brains, suggesting a potential feedback relationship between effective RBP concentrations and protein quality control in all ALS/FTD cases.

TAR DNA binding protein 43 (TDP-43), another member of the hnRNP family, possesses two RNA recognition motifs (RRMs), a nuclear localization signal (NLS), and a prion-like domain at the C-terminus [80]. Numerous mutations in TDP-43 have been associated with ALS and FTD [81,82]. The accumulation of TDP-43 is a major pathological feature of ALS and FTD [83,84,85], and inclusion bodies are observed in patients with abnormal expansions of r(GGGGCC)_n_, serving as a histopathological marker in 97% of ALS cases and 45% of FTD cases.

In contrast to hnRNP H, which directly associates with r(GGGGCC)_n_, the pathogenic mechanism of TDP-43 in ALS/FTD is believed to involve its interaction with DPRs, which are non-ATG translation products of r(GGGGCC)_n_ [15,86]. Edward B. Lee and colleagues discovered that DPRs induce TDP-43 protein lesions in an ALS/FTD model and trigger the onset and progression of FTD [81]. The amount and characteristics of produced DPRs, rather than the length of r(GGGGCC)_n_ repeats, determine the duration and severity of TDP-43 dysfunction.

### 3.2. FUS

Sarcoma fusion protein (FUS) is a 526-amino acid residue protein. [87] It is predominantly expressed in neurons and is involved in DNA and RNA metabolism through its interactions with motor proteins kinesin [88] and myosin-Va [89]. Missense mutations in the FUS gene have been associated with ALS [90,91], although the prevalence of FUS gene variants in the familial ALS population is low.

Sua Myong and colleagues conducted investigations on the binding of wild-type FUS to single-stranded RNAs, including r(GGGGCC)_4_, in a length-dependent manner. They observed the formation of a highly dynamic protein–RNA complex. The FUS–RNA interaction involves two mechanisms: (i) stable binding of FUS monomers to single-stranded RNA (ssRNA), and (ii) weak interaction of two FUS units with RNA, resulting in a highly dynamic interaction.

Higuro and workers observed the formation and phase transition of FUS condensates in vitro using purified full-length wild-type and mutant FUS proteins and r(GGGGCC)_4_. They found that FUS specifically forms complexes with r(GGGGCC)_4_ in a G4 structure-dependent manner, leading to a transition from liquid–liquid separation to liquid–solid transitions. Importantly, amino acid mutations associated with ALS significantly impact G4-dependent FUS condensation. These findings provide insights into the relationship between protein aggregation and dysfunction of FUS in ALS [49].

### 3.3. Zfp106

Zfp106 is a C2H2 zinc finger protein characterized by the presence of seven WD40 domains and four putative zinc fingers [92]. It plays a crucial role in maintaining neuromuscular signaling. Knockout mice exhibit gene expression patterns indicative of neuromuscular degeneration in their muscles and spinal cords. Interestingly, this phenotype can be reversed through motor neuron-specific repair of the Zfp106 transgene, highlighting its essential role in biological processes [93]. The functional acquisition model of *C9orf72* neurodegeneration has been investigated in a *Drosophila* model [94], where Zfp106 effectively mitigates the neurotoxicity associated with the expression of GGGGCC repeat in *C9orf72* ALS *Drosophila*. This suggests that Zfp106 acts as a repressor of neurodegeneration in *C9orf72* ALS models and demonstrates a functional interaction between Zfp106 and the r(GGGGCC)_n_ sequence. Furthermore, Brian L. Black and colleagues conducted pull-down assays and mobility shift assays, providing evidence that Zfp106 specifically binds to r(GGGGCC)_8_ but not to the sequence of r(AAAACC)_8_. The ability of Zfp106 to regulate normal cellular functions and inhibit ALS by binding to r(GGGGCC)_n_ makes it a potential drug target for treating ALS [45]. However, the mechanisms through which Zfp106 regulates normal cellular processes via RNA binding and how it inhibits ALS progression by interacting with r(GGGGCC)_n_ are still being investigated to guide drug design efforts [45].

### 3.4. ADARB2

ADARB2 is a member of the CNS-rich adenosine deaminase family, known for its role in mediating A-to-I (adenosine to inosine) editing of RNA [95]. It consists of two double-stranded-specific adenosine deaminase repeats, three double-stranded RNA-binding domains, and one editase domain spanning from the N- to C-terminus. The A-to-I editing activity primarily occurs within the 16–130 nucleotide interval. This enzyme selectively deaminates adenosine (A) residues in the double-stranded region of mRNA, converting them to inosine (I), which is recognized as guanine by the cellular translation machinery, resulting in codon alterations within the synthesized protein [46] (Figure 5).

Jeffrey D. Rothstein and colleagues conducted RNA fluorescence in situ hybridization (RNA FISH) and immunofluorescence labeling of RBP simultaneously in the induced pluripotent stem neuron (IPSN) cell line derived from *C9orf72*-related cases. Their study revealed the co-localization of ADARB2 protein with nuclear r(GGGGCC)_n_ RNA foci, while mRNA levels remained unchanged. Co-precipitation of ADARB2 with r(GGGGCC)_n_ repeats was also observed in vivo.

In vitro investigations utilizing recombinant ADARB2 through gel shift assays clearly demonstrated its binding to r(GGGGCC)_n_, implying the possible formation of ADARB2-RNA complexes. These collective findings indicate a strong binding between ADARB2 and r(GGGGCC)_n_. Furthermore, this team verified in vivo that the formation of r(GGGGCC)_n_ RNA foci requires the involvement of ADARB2 protein. Treatment of the IPSN line with specific siRNA targeting ADARB2 significantly reduced the number of RNA foci. However, further experimental evidence is still needed to fully elucidate ADARB2′s in vivo function [96]. Another unresolved aspect of ADARB2 function is the speculation that ADARB2 may lose its editing activity upon interaction with r(GGGGCC)_n_, although experimental validation of its downstream editing effects is currently lacking.

### 3.5. Purα

Pur-alpha (Purα) is a highly conserved DNA and RNA binding protein in eukaryotic cells [97]. It performs diverse physiological functions, including transcription activation or inhibition, cell growth, and translation [98,99]. While predominantly localized in the nucleus, Purα is also widely distributed in the cytoplasm of neurons, particularly in synaptic branches [88]. In the nucleus, Purα stimulates gene transcription by binding to mRNA transcripts and accompanying them to the cytoplasm. It remains associated with the mRNA during transport over considerable distances and functions at specific sites of mRNA translation [100]. The absence of Purα can lead to various neurological disorders [101,102].

The r(GGGGCC)_n_ repeat can sequester Purα, thereby impairing its normal functions such as gene transcription and mRNA translation, ultimately resulting in cell death [103]. In an ALS/FTD zebrafish model, Swinnen and colleagues demonstrated that the Pur2 domain of Purα binds to r(GGGGCC)_90_ repeat RNA [37]. Peng Jin and colleagues conducted studies on the pathogenesis of ALS/FTD, revealing that r(GGGGCC)_10_ can sequester Purα, a major component of RBPs, from the whole-cell lysate of mouse spinal cord [47]. Rossi and colleagues found that Purα can aggregate into cytosolic and nuclear granules in HeLa cells transiently transfected with a plasmid expressing r(GGGGCG)_31_. Nonetheless, due to the specific interaction between Purα and r(GGGGCC)_n_, it is conceivable that Purα may influence the outcome of RAN translation. Consequently, in ALS, reduced protein levels amplify certain cellular characteristics. Over-expression of Purα in mammalian and *Drosophila* model systems can rescue r(GGGGCC)_n_ repeat-induced neurodegeneration [47].

Furthermore, Purα also interacts with the C-terminal region of FUS, another protein recruited by r(GGGGCC)_n_ [104]. In vivo expression of Purα in various *Drosophila* tissues significantly exacerbates neurodegeneration caused by mutated FUS. Conversely, reducing Purα expression in neurons expressing mutated FUS significantly improves the climbing ability of *Drosophila* flies. This suggests that downregulation of Purα ameliorates locomotion defects, a classical symptom of ALS resulting from mutant FUS expression. These findings indicate that Purα may contribute to the pathogenesis of ALS mediated by FUS. However, it remains unclear which functional domains or subdomains of Purα are involved in mediating its interaction with FUS [105].

Binding of Purα to other cellular proteins can directly impact the expression of the *PURA* gene. Purα itself can bind to GC/GA-rich sequences in its own promoter and inhibit gene expression [106]. Similarly, binding of Purα to expanded polynucleotide repeat RNA may also affect the expression of the *PURA* gene. In both scenarios, the mechanism of action may involve the combination of Purα with cellular components, resulting in a reduction in effective intracellular Purα levels. The reduction in Purα could trigger a feedback mechanism of the *PURA* gene, although it is unknown whether this compensates for Purα sequestration [100].

## 4. Lead Small Molecules Binds to r(GGGGCC)_n_

Given the pharmacological advantages of r(GGGGCC)_n_ formation of RNA foci and their recruitment of RBPs, small molecules present an attractive option for targeting r(GGGGCC)_n_. Therefore, it is interesting to investigate the binding of r(GGGGCC)_n_ to small molecules (Figure 6). Currently, a number of the small molecules contain aromatic rings have been found to bind to r(GGGGCC)_n_.

### 4.1. Binding of r(GGGGCC)_8_ with the TMPyP4

The G4 structure has been shown to bind to 5,10,15,20-tetra(N-methyl-4-pyridyl) porphyrin (TMPyP4), as demonstrated before [107,108]. TMPyP4 binds a variety of G4 structures of DNA or RNA [109,110]. In 2014, Christopher E. Pearson and colleagues found that TMPyP4 could bind and distort the G4 formed by r(GGGGCC)_8_, inhibiting the interaction of some proteins with the repeat [23]. Several studies have shown that TMPyP4 disrupts the binding of hnRNPA1 to the r(GGGGCC)_8_ repeat, that are supposed to link to ALS/FTD pathogenesis [23]. Therefore, it may be possible to develop therapeutic treatments using TMPyP4 to disrupt the interaction of RBPs. However, TMPyP4 may either stabilize or destabilize RNA G4. Kelly and colleagues used molecule dynamics simulations to analyze RNA G4 structure and speculated that TMPyP4 might interact with RNA G4 in three different ways: top-stacking, bottom-stacking, and side-binding, maintaining stability under certain conditions [111]. However, the specific structure and binding mode of the complex have not been reported. Therefore, further study on the interaction between TMPyP4 and r(GGGGCC)_n_ RNA, as well as the destruction of RBPs binding which may cause toxicity, will be one of the directions for the development of related small molecule drugs.

### 4.2. Binding of r(GGGGCC)_8_ with Other Liands

Matthew D. Disney and colleagues has discovered three lead compounds, **1a**, **2**, and **3**, that bind with r(GGGGCC)_8_ in vitro, with Kds of 9.7, 10, and 16 μM, respectively [55]. These three small molecules were obtained by Hoechst or *bis-*benzimidazole query, and were derived from the small molecule library established by chemical similarity search. This library is enriched in compounds that have the potential to recognize RNA 1 × 1 nucleotide internal loops, among which 1a has been proven to bind 1 × 1 GG internal loops present in r(CGG)^exp^, and improve fragile X-associated tremor/ataxia syndrome (FXTAS)-associated defects [112].

As r(GGGGCC)_8_ RNA experiences dynamical equilibrium between hairpin and parallel G4 structure in solution, the binding constants of these lead compounds with RNA were evaluated in either K^+^ containing buffer (favorable for G4 structure) or Na^+^ buffer (favorable for hairpin). The 3–10 times higher Kds of **1a** and **3** were obtained in the presence of K^+^ than the Na^+^ buffer, demonstrating their favor binding to G4 structures of r(GGGGCC)_8_. In contrast, a Na^+^-dependent affinity of **2** was not affected by r(GGGGCC)_8_, but it significantly decreased with K^+^, showing the specific binding with hairpin structures. The optical melting data further demonstrated that compound **3** has no influence on the stability of r(GGGGCC)_8_, while compounds **1a** and **2** improve it.

The effects of three ligands on non-ATG translation of r(GGGGCC)_n_ were tested in HEK293 cells expressing r(GGGGCC)_66_ [55]. It was found that poly(GP) and poly(GA) proteins, but not poly(GR) proteins, were produced in the system. Compound **3** (100 µM, 24 h) was shown to moderately limit poly(GP) synthesis while having no effect on poly(GA). Compounds **1a** and **2**, on the other hand, drastically reduced the amounts of GP and GA proteins, which dramatically lowered the percentage of positive cells in the lesions. This suggests that ligand binding to r(GGGGCC)_n_ could be a potentially effective cure for FTD/ALS.

### 4.3. Binding of r(GGGGCC)_8_ with CB096

Disney and colleagues discovered a benzimidazole derivative CB096 that binds to r(GGGGCC)_n_. NMR, structure–activity relationship (SAR) studies, and molecular dynamics (MD) simulations with r(GGGGCC)_n_ hairpin structure have been used to determine the molecular interaction between CB096 and r(GGGGCC)_n_ (Figure 7) [113]. When r(GGGGCC)_n_ is folded, CB096 can specifically bind to the repeating 1 × 1 GG inner ring structure of 5′CGG\3′GGC. The TO-PRO-1 (TO-1) fluorescent dye replacement assay and microscale thermoelectrophoresis (MST) were used to screen the ligands bound to the r(GGGGCC)_8_ hairpin. CB096 binds to 5′CGG/3′GGC of the r(GGGGCC)_n_ hairpin and breaks the base pair as shown by NMR. To bind to the r(GGGGCC)_n_ hairpin structure, the chemical 5′s-NO2 group and 2-methoxyphenyl are crucial. In ALS/HEK293T FTD’s cells, CB096 slowed RAN translation and reduced poly(GP) DPR formation, but did not affect r(GGGGCC)_66_ mRNA levels. In conclusion, the researchers showed that CB096 binds particularly to the 1 × 1 GG inner ring 5′CGG\3′GGC generated during the expansion of r(GGGGCC)_n_.

### 4.4. Binding of r(GGGGCC)_n_ with DB1246, DB1247, and DB1273

Isaacs and colleagues screened a chemical library of small molecules to find the r(GGGGCC)_4_ binding ligands [53]. They identified 44 hits out of 138 small molecules by a FRET-based G4 melting assay. Among those hitting compounds, three molecules are structurally similar (DB1246, DB1247, and DB1273) and have the ability to bind and stabilize G4s structure, as shown by temperature dependent CD spectroscopy [53]. Treatment with these compounds led to a significant reduction in both RNA foci formation and dipeptide repeat protein levels in *Drosophila* carrying r(GGGGCC)_36_ and improved survival in vivo [53]. These findings suggest that targeting the r(GGGGCC)_n_ G4 using small molecules may be a promising therapeutic approach to alleviate two key pathologies associated with FTD/ALS.

### 4.5. Binding of r(GGGGCC)_n_ with CB253

Andrei and colleagues incorporated ^19^F modified nucleotides to replace the C6 residue in r(GGGGCC)_2_ duplex model (5′CCGGGG/3′GGGGCC) to investigate the binding mechanism of CB253 to r(GGGGCC)_n_ (Figure 8) [114]. The replacement of ^19^F nucleotide enables the use of ^19^F NMR spectroscopy to investigate the structure and interactions. Two types of inner ring, 1 × 1 GG and 2 × 2 GG, were detected and verified in the r(GGGGCC)_2_ hairpin structure. Among them, the 1 × 1 GG was the main conformation, and the two conformations could slowly transform into each other to achieve an equilibrium. Addition of CB253 stabilizes the 2 × 2 GG inner ring structure of r(GGGGCC)_2_ duplex, which becomes a stable dominant conformation. CB253 can form key interactions with N1-H of G3 and combine with r(GGGGCC)_2_ at a 2:1 ratio. The precise 2,4-diamino substitution pattern within CB253’s quinazoline scaffold is crucial for binding the r(GGGGCC)_n_ hairpin RNA. In HEK293T and lymphoblastoid cells from *C9orf72* patients, CB253 reduced the formation of stress granules induced by r(GGGGCC)_66_ and inhibited RAN translation in a dose-dependent manner, leading to a significant reduction in poly(GP) DPR levels. These findings indicate that CB253 is a promising chemical probe that can specifically bind to and stabilize the 2 × 2 GG inner ring of r(GGGGCC)_n_ hairpin structure, and inhibit various *C9orf72*-specific pathological mechanisms by directly engaging r(GGGGCC)_n_.

## 5. Summary and Perspective

In this review, we provide a comprehensive overview of the advancements in understanding the structure of r(GGGGCC)_n_ and d(GGGGCC)_n_, the phase separation and transition of r(GGGGCC)_n_, the interactions of r(GGGGCC)_n_ with RBPs, and the discovered ligands capable of inhibiting the non-ATG translations of r(GGGGCC)_n_ and/or the interactions between r(GGGGCC)_n_ and RBPs.

The relationship between the fatal neurodegenerative diseases ALS/FTD, the structure of r(GGGGCC)_n_ RNA, and their interactions have garnered significant research attention. When the repeat number exceeds the threshold, r(GGGGCC)_n_ RNA undergoes phase separation and transition, leading to the formation of nuclear RNA foci. These RNA foci recruit RBPs, disrupting the physiological functions of RNA splicing and maturation. Another pathogenic mechanism by which r(GGGGCC)_n_ contributes to ALS or FTD is the cytotoxicity of repetitive dipeptide proteins generated through non-ATG translation. Aggregates of these repetitive dipeptide proteins, can recruit numerous 26S proteasome complexes and stabilize a transient substrate-processing conformation of the 26S proteasome, suggesting impaired degradation processes [115].

Characterizing the repeat structure of r(GGGGCC)_n_ RNA and elucidating the structure-function relationship are key areas of research in understanding the pathogenic causes. r(GGGGCC)_n_ can adopt diverse structures, including hairpin and parallel G4 topologies, with equilibrium between them depending on solution conditions. However, the three-dimensional structures of r(GGGGCC)_n_ RNA are still unknown. Achieving a dominant conformation for structural studies may require sequence and solution condition optimization. Another challenging aspect is determining the secondary structures of r(GGGGCC)_n_ within RNA foci or gel-like states. Due to the non-crystalline solid state and heterogeneous nature of RNA foci, commonly used high-resolution structure determination methods such as X-ray crystallography or solution NMR are not applicable [116,117]. To date, the RNA structures within RNA foci remain unidentified. Advancements in RNA structure determination methodologies, such as solid-state NMR [118,119,120], are needed to overcome this limitation.

Several small molecules that bind to r(GGGGCC)_n_ have been discovered to block RBP interactions, inhibit phase separation, and/or hinder non-ATG translation, as evidenced both in vivo and in vitro. Understanding the structural details of the interactions between r(GGGGCC)_n_ RNA and ligands is crucial for facilitating the design of lead compounds to treat ALS/FTD. Similar to the challenges faced in studying r(GGGGCC)_n_ RNA, the complex structure determination of r(GGGGCC)_n_ RNA repeats and small molecules is lacking, necessitating further developments to gain insights into drug design.

Another known treatment approach for ALS/FTD involves the use of antisense RNA. Single-dose injections of antisense oligonucleotides (ASOs) targeting repeat-containing RNAs, while preserving mRNA levels encoding *C9orf72*, have resulted in sustained reductions in RNA foci and dipeptide-repeat proteins, leading to the amelioration of behavioral deficits. These efforts have identified the gain of toxicity as a central disease mechanism caused by repeat-expanded *C9orf72* and established the feasibility of ASO-mediated therapy [16]. ALS brains treated with ASO therapeutics targeting the *C9orf72* transcript or repeat expansion showed mitigation despite the presence of repeat-associated non-ATG translation products [46]. Moreover, the introduction of mRNA that encodes r(GGGGCC)_n_ binding proteins into ALS/FTD cells has the potential to restore RBP functions by augmenting the intracellular pool of RBPs recruited by RNA foci. This approach represents an alternative strategy for treating ALS by targeting r(GGGGCC)_n_ RNA. Lastly, gene editing system by CRISPR/Cas9 has successfully removed the GGGGCC repeat expansion in *C9orf72*, leading to reduction in RNA foci and DPR formations, proving a promising approach in ALS treatments.

## Figures and Tables

**Figure 1 molecules-28-05801-f001:**
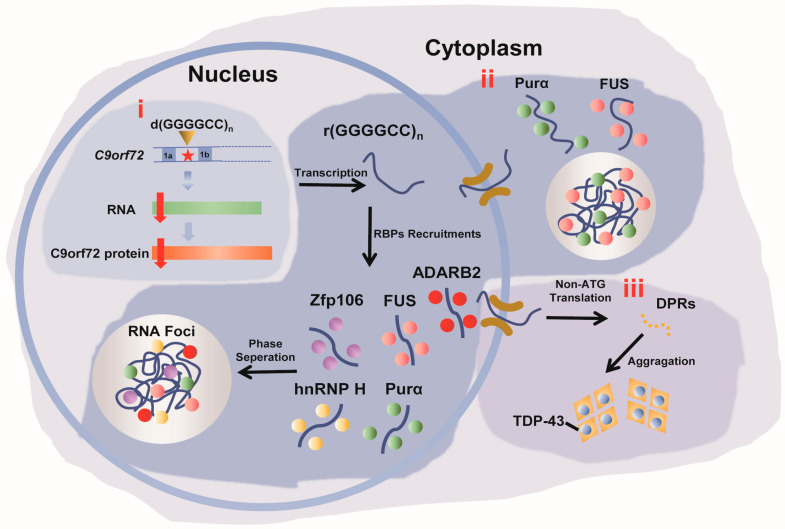
Three pathogenic mechanisms associated with *C9orf72*-related FTD and ALS. (**i**) Aberrant expansion of the d(GGGGCC)n of *C9orf72* within contains two non-coding exons (1a and 1b), and under pathological conditions, repress transcription, resulting in the reduction in *C9orf72* protein. (**ii**) The transcribed r(GGGGCC)n aggregates in the nucleus to form RNA foci that recruit RBPs, affecting the intra cellular functions of RBPs, i.e., splicing. (**iii**) The r(GGGGCC)n RNA is transported into the cytoplasm and undergoes repeat-associated non-ATG translation, resulting in the synthesis of DPRs. The DPRs forms aggregation and associate TDP-43, which induce cytotoxic effects in cells. The labels (**i**–**iii**) correspond to the three pathological mechanisms. The red star highlights the hexanucleotide repeat GGGGCC in the non-coding region of *C9orf72*.

**Figure 3 molecules-28-05801-f003:**
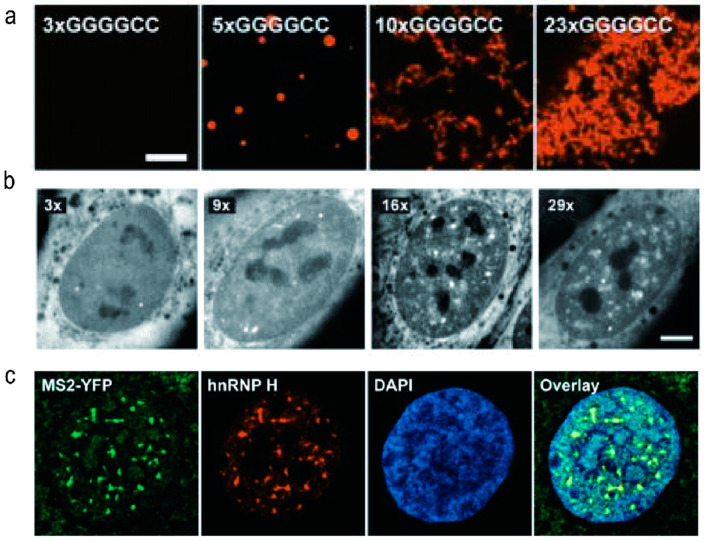
Representative results of biological phase separation of r(GGGGCC)_n_. (**a**) The in vitro fluorescence imaging of r(GGGCC)_n_ RNA clusters at indicated number of r(GGGGCC)_n_. (**b**) Representative fluorescence micrographs and corresponding quantification of the total volume of foci per cell in U-2OS cells transduced with r(GGGGCC)_n_ RNA with the indicated number of r(GGGGCC)_n_. (**c**) Representative immuno fluorescence images illustrating that the r(GGGGCC)_29_ recruited endogenous hnRNP H. Figure 3 was reprinted from the reference [43].

**Figure 4 molecules-28-05801-f004:**
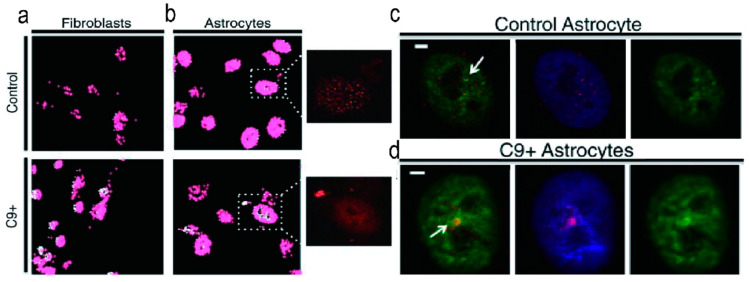
Quantification of stained BG4-foci and area was performed in fibroblasts and astrocytes derived from patients with ALS/FTD and healthy controls. Representative images of non-ALS and ALS fibroblasts (**a**) and astrocytes (**b**) are shown, with the ‘BG4 Count’ projection representing all stained areas above the determined threshold (showed in red), and areas of particularly dense staining shown in white. The inset displays the source image, highlighting only the red channel corresponding to BG4-FLAG staining. (**c**) Control astrocytes exhibit single, small nuclear hnRNP H/BG4 foci. (**d**) Patient astrocytes demonstrate nuclear hnRNP H/BG4 foci. Figure 4 was reprinted from the reference [79].

**Figure 5 molecules-28-05801-f005:**
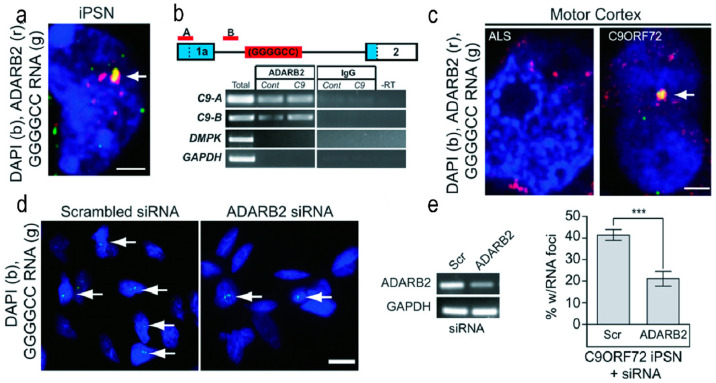
ADARB2 Protein Binds to the r(GGGGCC)_n_. (**a**) Colocalization of r(GGGGCC)_n_ RNA foci with ADARB2 signal in IPSN cells. (**b**) Co-immunoprecipitation (co-IP) of ADARB2-bound RNA isolated from control and *C9orf72*-induced cell lines. RT-PCR of the co-IP RNA using two primer sets (A and B, red), located upstream of the r(GGGGCC)_n_ repeat, demonstrated ADARB2 binding to *C9orf72* RNA in both control and *C9orf72* cell lines. (**c**) Colocalization of r(GGGGCC)_n_ RNA foci and ADARB2 was observed in postmortem motor cortex tissue from *C9orf72* patients. (**d**,**e**) Knockdown of ADARB2 using siRNA significantly reduced the percentage of nuclear RNA foci (indicated by arrows). siRNA knockdown of ADARB2 results in a significant reduction in the percent of iPSNs with nuclear RNA foci (arrows). Data in (E) indicate mean ±SEM (*** *p* < 0.001). Figure 5 was reprinted from the reference [46].

**Figure 6 molecules-28-05801-f006:**
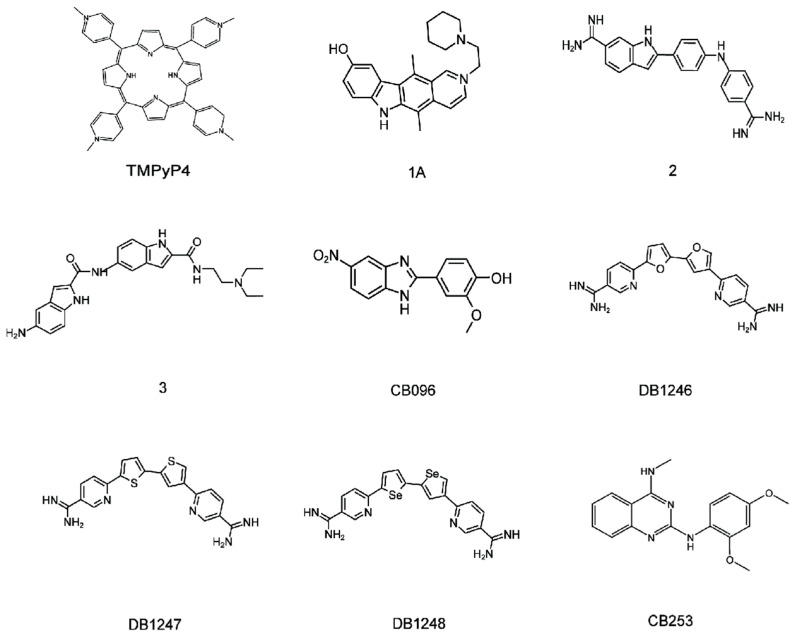
r(GGGGCC)_n_ small molecular structure bound to small molecules.

**Figure 7 molecules-28-05801-f007:**
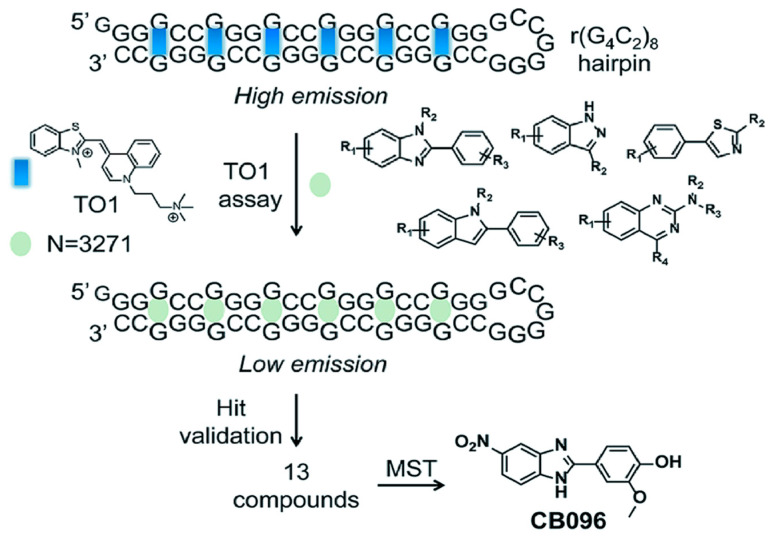
CB096 specifically bound to the repeated 1 × 1 GG inner loop structure 5′CGG/3′GGC in the r(GGGGCC)_8_ hairpin structures. Figure 7 was reprinted from the reference [113].

**Figure 8 molecules-28-05801-f008:**
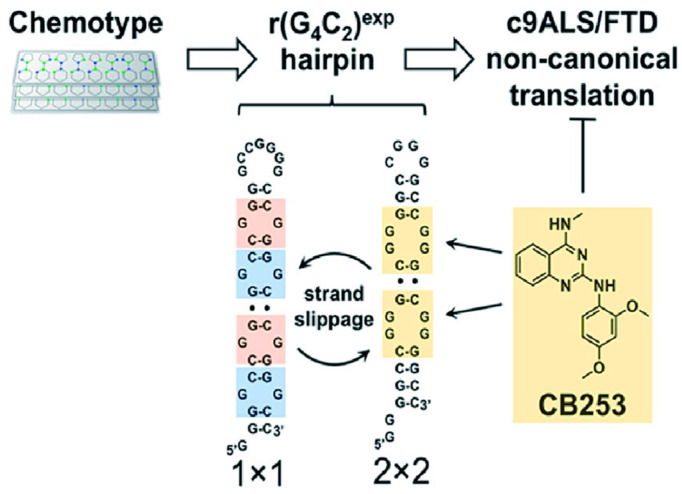
CB253 that selectively binds the hairpin form of r(GGGGCC)_n_. Figure 8 was reprinted from the reference [114].

## Data Availability

Not applicable.

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
