# Peer review of "Advances in the Structure of GGGGCC Repeat RNA Sequence and Its Interaction with Small Molecules and Protein Partners"

_molecules, 2023, doi:10.3390/molecules28155801_

Round 1

Reviewer 1 Report

Overall this is a very well written, focused, and thorough review of the associated GGGCC repeat expansion in C9orf72 and ALS/FTD. The structural biology of GGGCC expansion in the underlying disease pathogenesis is quite interesting. The review continues with the hypothesis that RNA foci/RNABP aggregation as the root of pathogenesis and provides significant evidence in support, providing a number of examples of RNABPs involved in pathology. The Authors conclude with potential small molecule or antisense RNA pharmacological interventions targeting the disruption of the RNA foci/RNABP aggregation. Unfortunately, the authors neglected to include what most likely will be the ultimate treatment, CRISPR/Cas9-HDR as demonstrated by Meijboom et al. Nature Communications 2022

A Brief description of some of the other etiologies (eg SOD1, TARDBP mutations, idiopathic) would be helpful to prevent a novice reader from concluding the GGGCC expansion of C9orf72 was the definitive causal mutation.

Additionally, for the novice's benefit (which is the primary target of all good reviews) a brief introduction to microsatellite repeat expansion disorders would provide historical context and allow comparisons of the varying pathologies. The identified repeat expansion in C9orf72 is rather recent compared to those found earlier (myotonic dystrophy, fragile X, Huntington's)

Minor Typo

Pg 6, Line 198 "Diseases related RBPs…" should be "Disease related RBPs"

Reviewer 2 Report

Overall, this review does a great job by summarizing the advances in the study of disease relevant r(GGGGCC)n. The authors introduced the structure of GGGGCC RNA repeats and the interaction with related proteins within the RNA foci. Also, several compounds that target such RNA structure have also been introduced, which provides a perspective for RNA-targeted therapy. So, this work is worth of publication in Molecules. However, the following concerns should be addressed before publication.

1. Normally, G-quadruplex is abbreviated as G4 instead of G-Q.

2. Figure 1 nicely summarize the three pathogenic mechanisms associated with GGGGCC repeat. Please indicate the corresponding mechanism by labeling in the figure (adding i, ii, iii).

3. In Figure 2a, the spelling of the word “G-quadruplex” is not correct. If it is from the original paper, please correct it by refining the label.

4. In the beginning of 2.3, by introducing liquid-liquid phase separation (LLP), one or two references about LLP can be added, which gives an overview of the biological significance.

5. Section 3.1, there should be reference added after the first sentence when introducing hnRNP H and TDP-43.

6. Section 4.2, when the authors introduce the work from Disney lab, there should be reference regarding the 1A, 2, and 3 compounds. And are the Kds in mM? or µM/nM range? The structures look similar to Hoechst dye, which should be tight and non-specific binders.

Minor editing of English language is required.
